# A Projection Method for the Estimation of Error Covariance Matrices for Variational Data Assimilation in Ocean Modelling

**Jose M. Gonzalez-Ondina** [1,*] **, Lewis Sampson** [2] **and Georgy I. Shapiro** [3]

1 University of Plymouth Enterprise Ltd. (UoPEL), Drake Circus, Plymouth PL4 8AA, UK
2 Met Office, FitzRoy Road, Exeter EX1 3PB, UK; lewis.sampson@metoffice.gov.uk
3 School of Biological and Marine Sciences, University of Plymouth, Drake Circus, Plymouth PL4 8AA, UK; g.shapiro@plymouth.ac.uk
* Correspondence: jose.ondina@plymouth.ac.uk

**Abstract:** Data assimilation methods are an invaluable tool for operational ocean models. These methods are often based on a variational approach and require the knowledge of the spatial covariances of the background errors (differences between the numerical model and the true values) and the observation errors (differences between true and measured values). Since the true values are never known in practice, the error covariance matrices containing values of the covariance functions at different locations, are estimated approximately. Several methods have been devised to compute these matrices, one of the most widely used is the one developed by Hollingsworth and Lönnberg (H-L). This method requires to bin (combine) the data points separated by similar distances, compute covariances in each bin and then to find a best fit covariance function. While being a helpful tool, the H-L method has its limitations. We have developed a new mathematical method for computing the background and observation error covariance functions and therefore the error covariance matrices. The method uses functional analysis which allows to overcome some shortcomings of the H-L method, for example, the assumption of statistical isotropy. It also eliminates the intermediate steps used in the H-L method such as binning the innovations (differences between observations and the model), and the computation of innovation covariances for each bin, before the best-fit curve can be found. We show that the new method works in situations where the standard H-L method experiences difficulties, especially when observations are scarce. It gives a better estimate than the H-L in a synthetic idealised case where the true covariance function is known. We also demonstrate that in many cases the new method allows to use the separable convolution mathematical algorithm to increase the computational speed significantly, up to an order of magnitude. The Projection Method (PROM) also allows computing 2D and 3D covariance functions in addition to the standard 1D case.

**Keywords:** data assimilation; variational methods; analysis of innovations; ocean modelling; operational forecast

## 1. Introduction

Due to intrinsic inaccuracies in the model equations, numerical schemes and quality of input data streams, even the best ocean models gradually deviate from reality and can only be considered an estimate of the true ocean state [1]. The introduction of Data Assimilation (DA) techniques allowed to reduce the deviation of models from the true state, vastly improving the accuracy of ocean forecasting [2].

There are a number of DA methods currently in use [1,3]. Many modern operational Variational Data Assimilation schemes have originated from the method of optimal interpolation Gandin [4,5], and are based on minimising the so-called cost function, see, e.g., Lorenc [6], Waters et al. [7] and Carrassi et al. [1]. The cost function includes a combination of the model forecasts and the observational data, weighted by the relative correctness

of each component as represented by their error covariance matrices (ECM). This DA technique allows to compute a more accurate state called the analysis, which then is used as an initial condition for a new forecasting cycle.

The cost function used to obtain the analysis vector with the maximum likelihood of concordance with the truth [2] is given below (see, e.g., [8])

$$J[\delta x] = \delta x^T \mathbf{B}^{-1} \delta x + (H(\delta x) - d)^T \mathbf{R}^{-1}(H(\delta x) - d), \tag{1}$$

$$\delta x = x^a - x^b, \quad d = y^o - H\left(x^b\right), \tag{2}$$

where observational and background data are represented by vectors $y^o$ and $x^b$, respectively. Vector $d$ contains the so-called innovations (differences between observation values and model values interpolated at the observation location), $\delta x$ the analysis increment, and $x^a$ the analysis vector. The operator $H$, referred to as the observation operator, takes model data into observational space. The matrices $\mathbf{B}$ and $\mathbf{R}$ are the ECMs for background and observational data, respectively (see Ide et al. [9], for a description of the notation).

The calculation of the background (or forecast) and observation error covariance matrices (ECM) is a key component of variational data assimilation systems. The more precisely they are estimated, the smaller difference there is between the analysis fields and the true state of the ocean [10]. In practice, the estimation of the ECMs is not an easy task. Firstly, observations are usually scarce. Even today, when satellite imagery is at one's disposal, only some data are available in large quantities and only at the ocean surface. Data within the water column are limited to places where buoys or drifters exist. Secondly, the true state of the ocean is never known, so the methods rely on statistics on the available observations to estimate it. Errors has to be estimated by some proxy, typically either from innovations or model differences [10]. Thirdly, ECM estimation requires to process large amounts of data coming from the model and the observations. The computation of all correlations from a set of $N$ observations requires to carry out $\mathcal{O}(N^2)$ floating point operations. This is a mammoth task even for modern high-performance computers.

Since there are such computational difficulties with calculating the ECMs, simplifications have been developed in order to make the process feasible. These include parametrising the covariance matrices [7] and the assumption of isotropic error correlation [11,12]. The isotropic assumption means that the direction is ignored when calculating correlation, and distance becomes the only determining variable for the correlation. A widely used method for computing the error covariances under assumptions of statistical isotropy and lack of spatial correlation between observational errors was designed by Hollingsworth and Lönnberg [13] and is hereafter called the H-L method, which is used in many operational systems (see, e.g., [1]). Due to the nature observations it is common to assume that they are spatially uncorrelated [14], and therefore allow $\mathbf{R}$ to be a diagonal ECM. Using this assumption, the H-L method is able to produce a joint estimate for both the observation and background EMCs.

The H-L method produces the covariance function for each model grid node using the following steps [13]. First, all possible products of pairs of innovations relevant to each model grid node, are calculated and binned by their distances. Second, the products related to a certain bin are averaged (usually using time averaging) to give an approximate value of covariance. Third, a curve is fitted through the covariance points (not taking into account the first bin) to produce a covariance function which then is used to calculate the diagonal element of $\mathbf{R}$ related to that grid node, and a row of elements for the $\mathbf{B}$ matrix, both diagonal and non-diagonal. Although the binning step was not considered an essential part of the H-L method by the authors, it was recommended when the amount of data is large. In modern practice, the number of innovations can be in the order of millions and, more importantly, observations are rarely taken at the same locations, therefore, binning is unavoidable (see, e.g., [15,16]). Despite being widely used, the H-L method has some weaknesses such as inability to include anisotropy and underperformance in sparsely observed areas.

The proposed Projection Method (PROM) follows the general variational approach based on minimisation of the cost function and aims to improve upon the H-L method. Instead of calculating the covariances in bins and then fitting a curve to them, PROM uses exact locations of the observations and fits the covariance curve using all individual products of innovations. The PROM method does not require the assumption of isotropy as it always finds the best fitting surface instead of a curve.

Section 2 presents a description of the method and the data used for the numerical experiments, Section 3 presents the results of numerical experiments using PROM for two cases from an operational model of the North Indian Sea. One case considers a slightly idealised situation when the true covariance function is known, and the second relates to the actual outputs from the North Indian Sea model. Performance of PROM in comparison to the H-L method is considered in the Section 4. The Section 5 gives a final summary of the main points presented in the paper.

## 2. Materials and Methods

We have developed a method to jointly estimate covariance matrices **B** and **R** based on the analysis of innovations. This method is similar in concept to H-L, but it removes the intermediate step of binning the innovations. As we shall see, removing this step requires to look at the analysis of innovations and function fitting from a different point of view. The following subsections describe the PROM method in detail.

### 2.1. Analysis of Innovations

The core concept of the analysis of innovations is that the model error covariances can be computed from the innovation covariances:

$$\jmath = x^t - x^b, \quad \epsilon = x^t - y^o, \tag{3}$$

$$d = y^o - H\left(x^b\right) = (x^t + \epsilon) - (x^t + \eta) = \epsilon - \eta, \tag{4}$$

where $\eta$ and $\epsilon$ are the background and observation errors, respectively. To calculate the covariance of the innovations, some assumptions are made. The first assumption is that the errors are not biased, which simplifies the equation of the covariance for innovations to:

$$f(r) = \overline{d(r_0)d(r)}, \tag{5}$$

where $d(r) = \epsilon(r) - \eta(r)$ is a random variable that represents the innovations at $r$, $f(r)$ is the spatial covariance of innovations and $r$ is the 2- or 3-dimensional relative position vector from a grid point assumed to be located at $r_0$. The over-line means ensemble average but, in practice, the ergodic hypothesis is invoked allowing to replace the ensemble average with a time average over a period when the process can be considered statistically in a steady state [17].

By substituting the definition of $d$ into Equation (5) we obtain:

$$\begin{aligned} f(r) &= \overline{(\epsilon(r_0) - \eta(r_0))(\epsilon(r) - \eta(r))} \\ &= \overline{\epsilon(r_0)\epsilon(r)} - \overline{\eta(r_0)\epsilon(r)} - \overline{\epsilon(r_0)\eta(r)} + \overline{\eta(r_0)\eta(r)}. \end{aligned} \tag{6}$$

Assuming that observations and forecast error are uncorrelated, and that for nonzero separation the observation errors are also uncorrelated, the innovation covariance can be written as follows:

$$f(r) = \begin{cases} \overline{\epsilon(r_0)\epsilon(r_0)} + \overline{\eta(r_0)\eta(r_0)}, & \text{for } r = r_0 \\ \overline{\eta(r_0)\eta(r)}, & \text{for } r \neq r_0. \end{cases} \tag{7}$$

As the number of observations increase the innovations statistics converge in Equation (5) to a function for forecast error covariance for $r \neq r_0$. At $r = r_0$, $f(r)$ is not

continuous, because error variances include both forecast and observation errors [10,13] and for this reason, this point is treated differently in all methods based on innovations.

### 2.2. Fitting the Covariance Model to Innovations

The main idea of the PROM method is that the binning step used in H-L method is unnecessary, except for a "central bin" located at grid point $r_0$. This is to cover the situation in which the actual observations near $r_0$ are located at slightly different locations, but the covariance function will be computed at $r_0$. For illustration, let us consider the 2D case shown in the sketch in Figure 1 on page 4, where the isotropy assumption is not used. The red/blue dots, located at $\Delta r = r_j - r_i$, are all the products of pairs of innovations $d(r_i)d(r_j)$ for which $r_i$ is inside the central bin of the target point $r_0$, defined as all the points at a small distance $L_c$ from $r_0$. Interestingly, if a surface (e.g., a 2D Gaussian) is fitted to these products, it will be a good approximation to the true covariance function. In what follows we will prove why fitting to the products produces a good estimate of the covariance and we will show how to compute this fit efficiently.

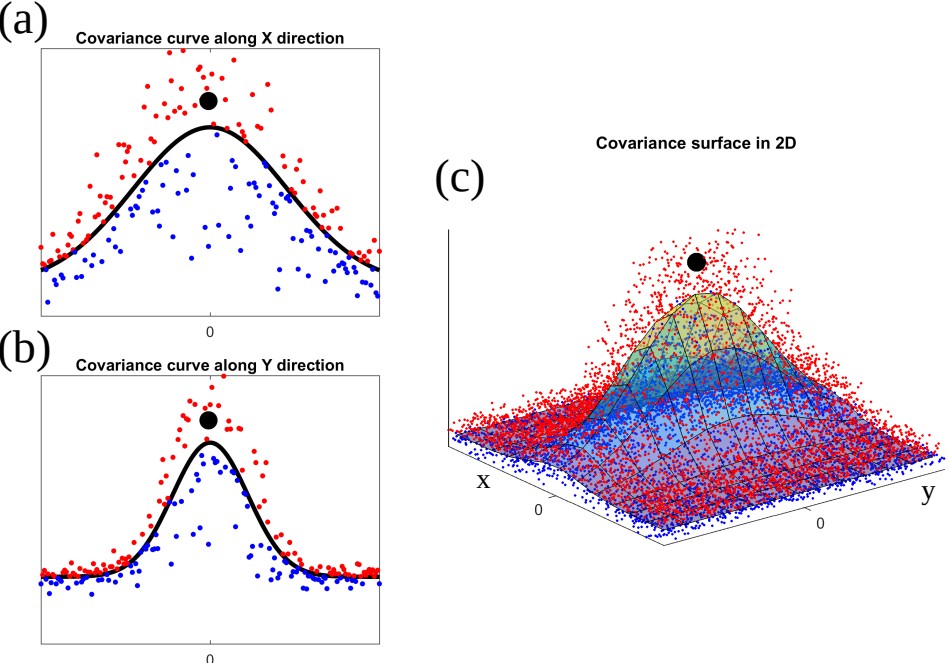

**Figure 1.** Sketch of the PROM method for a 2D case (i.e., anisotropic). The dots are products of pairs innovations (where one of the elements of the pair being close to the point where the covariance is computed) located at position $\Delta r$ computed as the vector difference between the locations of the "far" innovation and the one in the central bin. The model covariance (an anisotropic 2D Gaussian surface in this case) is fitted to all innovation products. For clarity we have plotted in red colour the products above the fitted Gaussian surface and in blue those below, representing the fact that the fitted covariance function tends to be somewhere in the middle. The plots in panels (**a**,**b**) represent profiles of the covariance functions along the *X* and *Y* axis to show the anisotropy of the function and data while the plot in panel (**c**) shows the full covariance function fitted to all the data. The black dot represents the variance of the innovations at the centre $r_0$.

From a mathematical point of view, the PROM method defines a subspace of functions (a parametric model of the covariance) and then finds the function with minimum distance to the data *projected* into that subspace. The method also assumes that the background model and observations are unbiased and the local homogeneity of the covariance, as it is done for the H-L method. The subspace can be defined by any basis of functions, including anisotropic ones.

Let us assume that, for certain physical variable, we have a set of $N$ innovations $d_i$, $i = 1, \ldots N$ computed as differences of observations performed at locations $r_i$ and times $t_i$ and a background model interpolated at $(r_i, t_i)$. All innovations are contained in the spatial domain and time range of interest. The times are assumed to be multiple of some interval: $t_i = k\Delta t$ (for example, $\Delta t$ is equal to one day) for $k = 1 \ldots T$. To simplify the following derivation, we will define $f_r$ to be the covariance function centred at $r$ as (see Equation (7)):

$$f_r(\Delta r) = \overline{d(r,t)d(r + \Delta r, t)}, \quad \text{for } r \neq 0, \tag{8}$$

where $r \in \Omega$ is any location in the domain, $\Omega$, $\Delta r$ is a relative location to $r$, $t$ is time and the over-line stands for the time average, which by invoking the ergodic hypothesis we use in place of the ensemble average. In practice, $f_r$ is never computed exactly, instead it is approximated by a simple model $\tilde{f}_r(\Delta r, a_0, a_1, \ldots)$, where $a_i$ are parameters that, in general, depend on $r$.

In order to obtain the values of the model parameters $a_i$ that better approximate $\tilde{f}_r$ to $f_r$, the first step is to define what we mean by "approximate", that is, a measure of how close a function is to another. For this we will define a $\|\bullet\|$ that allows to compute a distance between two functions $\|f_{r,t} - \tilde{f}_{r,t}\|$. The definition of this norm is the most important aspect of the method because it determines how faithful the model will be to the true covariance, and how difficult the computation of the model will be. The next subsection will be devoted to defining this norm. Once this is done, the fitting parameters $a_i$ will be computed using a norm minimisation approach similar to the least squares method.

### 2.3. Distance between Functions

The only points were we have information about innovations are the spatial locations where the observations were made. For this reason we chose the following norm defined for functions $f_r \in V$, where $V$ is the space of all real functions on the domain $\Omega$ with origin at $r$:

$$\|f_r\| \equiv \sqrt{\sum_{i=1}^{N}[f_r(\Delta r_i)]^2}, \tag{9}$$

where $\Delta r_i = r_i - r$. This discrete norm takes into account the values of the function at all locations where the observations were made. It can be thought as a "sampling" norm, where the values of the function are only picked at certain locations. This type of norm is a good choice if the locations $r_i$ are scattered more or less uniformly in the domain, which is true in most practical cases.

This norm can be derived from an inner product as the canonical norm for said inner product: $\|f_r\| = \sqrt{\langle f_r, f_r \rangle}$, where:

$$\langle f, g \rangle = \sum_{i=1}^{N} f(r_i)g(r_i). \tag{10}$$

Assuming that functions are identified if and only if they coincide on all the points $r_i$, it is trivial to check that the definitions above satisfy the conditions for norms and inner products for the space of functions $V$.

However, there is still one difficulty. The covariance functions $f_r$ are not known, not even at points $r_i$, only the innovations $d(r_i, t_i)$ are. This means that norms and inner products including $f_r$ cannot be computed exactly, they have to be approximated. This will be described in Section 2.5, meanwhile, we will assume that we can compute any norm or inner product.

### 2.4. Finding the Projection

We search for a function $\tilde{f}_r$ as close as possible to the true $f_r$. In other words, we search for:

$$\min_{a_1,a_2,\ldots a_m} \left\| \tilde{f}_{\boldsymbol{r}}(\Delta \boldsymbol{r}, a_1, a_2, \ldots a_m) - f_{\boldsymbol{r}} \right\| \tag{11}$$

Let us assume that all $\tilde{f}_{\boldsymbol{r}}$ conform a finite dimensional function subspace $V^m \subset V$, so we can write:

$$\tilde{f}_{\boldsymbol{r}}(\Delta \boldsymbol{r}, a_1, a_2, \ldots) = a_1 \phi_1(\Delta \boldsymbol{r}) + a_2 \phi_2(\Delta \boldsymbol{r}) + \ldots + a_m \phi_m(\Delta \boldsymbol{r}), \tag{12}$$

where $\phi_0$, $\phi_1$, ... are a basis of $V^m$, the $m$-dimensional space of all the Equations $\tilde{f}_{\boldsymbol{r}}$. (11) is equivalent to:

$$\min_{a_1,a_2,\ldots a_m} \left\langle \tilde{f}_{\boldsymbol{r}}(\Delta \boldsymbol{r}, a_1, a_2, \ldots a_m) - f_{\boldsymbol{r}}, \tilde{f}_{\boldsymbol{r}}(\Delta \boldsymbol{r}, a_1, a_2, \ldots a_m) - f_{\boldsymbol{r}} \right\rangle, \tag{13}$$

which can be computed by solving the system resulting from differentiating the inner product with respect to each coefficient $a_i$ and then equalling to zero. After simplification this system can be written as:

$$\left\langle \tilde{f}_{\boldsymbol{r}}(\Delta \boldsymbol{r}, a_1, a_2, \ldots a_m), \phi_j \right\rangle = \left\langle f_{\boldsymbol{r}}, \phi_j \right\rangle, \quad \forall j = 1, 2, \ldots m, \tag{14}$$

which is equivalent to finding the projection of $f_{\boldsymbol{r}}$ onto $V^m$.

For simplicity, in what follows we will consider $m = 2$, so there are only two basis functions $\phi_1$ and $\phi_2$, but the process is exactly the same for the case of more functions. Substituting $\tilde{f}_{\boldsymbol{r}}$ by $a_1 \phi_1(\Delta \boldsymbol{r}) + a_2 \phi_2(\Delta \boldsymbol{r})$:

$$\begin{aligned} \langle a_1 \phi_1 + a_2 \phi_2, \phi_1 \rangle &= \langle f_{\boldsymbol{r}}, \phi_1 \rangle \\ \langle a_1 \phi_1 + a_2 \phi_2, \phi_2 \rangle &= \langle f_{\boldsymbol{r}}, \phi_2 \rangle. \end{aligned} \tag{15}$$

Simplifying by using the linearity of the inner product:

$$\begin{aligned} a_1 \langle \phi_1, \phi_1 \rangle + a_2 \langle \phi_2, \phi_1 \rangle &= \langle f_{\boldsymbol{r}}, \phi_1 \rangle \\ a_1 \langle \phi_1, \phi_2 \rangle + a_2 \langle \phi_2, \phi_2 \rangle &= \langle f_{\boldsymbol{r}}, \phi_2 \rangle. \end{aligned} \tag{16}$$

Now, under the assumption that all the inner products in the system of equations above can be computed, the system can be solved for the coefficients $a_1$ and $a_2$. The resulting function $\tilde{f}_{\boldsymbol{r}}$ will be the one in $V^2$ closest to $f_{\boldsymbol{r}}$ in terms of the discrete norm.

The system of Equation (16) must be solved for all points for which we want to estimate the covariance. The coefficients $a_1$ and $a_2$ depend on $\boldsymbol{r}$ because the right-hand side of Equation (16) depends on $\boldsymbol{r}$. The inner products can be computed very quickly and efficiently in modern hardware and the system of two linear equations can be solved using any simple method as it is a very quick operation. Even in more general cases, the number of unknowns $a_i$ is small and the typical system is well-conditioned. Only when the amount of available data for a particular node is very small, there may be cases where the system is ill-conditioned. These cases can be detected (for example, by looking at the number of available data or the condition number of the equation matrix) and removed from the final results.

Once the projection is found, matrix **B** can be parametrised using $a_1$ and $a_2$ [18]. Matrix **R** can then be computed by subtracting $\tilde{f}_{\boldsymbol{r}}(\boldsymbol{0}) = a_1 + a_2$ from the variance at $\boldsymbol{r}$ computed in the central bin (see Equation (7)). This is completely analogous to what is done in the H-L method.

### 2.5. Approximating the Inner Product

There is still one difficulty—how to compute the inner products $\langle f_{\boldsymbol{r}}, \phi \rangle$. Let us start by substituting the definition of the inner product and $f_{\boldsymbol{r}}$:

$$\langle f_{\boldsymbol{r}}, \phi \rangle = \sum_{i=1}^{N} f_{\boldsymbol{r}}(\Delta \boldsymbol{r}_i) \phi(\Delta \boldsymbol{r}_i) = \sum_{i=1}^{N} \overline{d(\boldsymbol{r}, t) d(\boldsymbol{r} + \Delta \boldsymbol{r}, t)} \phi(\Delta \boldsymbol{r}_i). \tag{17}$$

This expression is not computable because the innovations are known only at the points and times of observations, and not at any point $d(\mathbf{r}, t)$; however, this expression can be approximated by using the Central Limit Theorem. This theorem states that we can substitute each mean in a sum by one random value following the same distribution and the value of the sum will still converge to the original value. This can be done by assuming that choosing the particular time when the $i$-th observation was taken is a random choice.

$$\langle f_{\mathbf{r}}, \phi \rangle = \sum_{i=1}^{N} d(\mathbf{r}, t_i) d(\mathbf{r}_i, t_i) \phi(\Delta \mathbf{r}_i) + O\left(N^{-1/2}\right). \tag{18}$$

The value of the innovation $d(\mathbf{r}, t_i)$ is still not known. By using the hypothesis of local homogeneity we can approximate it as:

$$\langle f_{\mathbf{r}}, \phi \rangle = \sum_{i=1}^{N} d_0(t_i) d(\mathbf{r}_i, t_i) \phi(\Delta \mathbf{r}_i) + O\left(N^{-1/2}\right), \tag{19}$$

$$\text{for } D_0(t_i) = \frac{1}{n_i} \sum_{j=1}^{n_i} d\left(\mathbf{r}_{k(i,j)}, t_{k(i,j)}\right),$$

where $\Delta \mathbf{r}_i = \mathbf{r}_i - \mathbf{r}$; $k(i, j)$ the indices of the $j = 1, ..., n_i$ innovations inside the central bin at time $t_i$ and $d\left(\mathbf{r}_{k(i,j)}, t_{k(i,j)}\right)$ are these innovations; in other words, those such that $\|\mathbf{r}_j - \mathbf{r}\| < L_c$, for some distance $L_c$. This distance $L_c$ is usually chosen as half the size of the output grid cell. This approximation of the inner product converges fast when the number of innovations $N$ is large enough. In contrast to the H-L method, which only uses one data point (the covariance) in each bin, the PROM method considers all the observations (and hence innovations) as if they were taken at the same time.

### 2.6. Data

To test the PROM method and perform comparisons with H-L we have used real innovations computed in an ocean operational model of the North Indian Sea [19]. This area was chosen because its circulation pattern is complex, as it is its bathymetry, containing areas of deep and shallow waters. This model uses NEMO v3.6 stable [20] for the dynamic simulation and NEMOVAR [21] as the DA engine. The computational mesh for NEMO uses hybrid enveloped sigma coordinate system [22] (see Figure 2 on page 9).

As it was done for the operational model, for this paper we have assumed that the covariances have seasonal variability, but that they can be considered statistically uniform during a season for the purpose of computing the error covariances. The innovation data we used were already preprocessed, as part of the operational system, to remove biases from observations and model and passed a quality control that removed the erroneous observations. We chose to use sea surface temperature (SST) because it had the largest amount of data available.

For some tests we used what we call "idealised" or "synthetic" data. The difference between "real" and "idealised" cases are explained in the following sections. These idealised data allow to test the PROM and H-L methods in controlled conditions of data spatial and time density, but knowing exactly what the true covariance function is. The way the synthetic innovation data are generated is described in Section 3.1. The "real" case uses the actual innovations supplied to the operational model for data assimilation.

## 3. Results

To test the method we have used data as realistic as possible to test the characteristics of our method. Firstly, we introduce an "idealised case" which uses the real location of real innovations for a domain in the North Indian Sea. The "idealised" part comes from the fact that, instead of using actual innovations for which the covariance function is unknown, we

use a function with known correlation. This allows to compare the skills of both methods in a well-known scenario.

Secondly, we present a comparison with real temperature innovations. In this case the covariance is not known a priori, but the quality of the results obtained, combined with the knowledge obtained from the idealised case, allows to confirm the benefits of the proposed method.

### 3.1. Idealised Case

For this case we used the location and time of Sea Surface Temperature (SST) innovation data for one year's summer season (JJA) to mimic a seasonal time-average. Instead of the actual innovation data we associated to these locations synthetic data of known spatial correlation. This part of the data is not a perfect representation of real innovations, but it will be a good test of the skill of the methods. If a method does not perform well in this situation it is not likely to do better in a real situation. Additionally, the performance of the PROM method relative to the density of data can be analysed.

The synthetic innovation data used follow the expression:

$$d_s(\boldsymbol{r}, t) = a_t \, e^{\|r - r_0\|^2 / 2b_0^2} + c\varepsilon_{\boldsymbol{r},t}, \tag{20}$$

where $\boldsymbol{r}_0$ is a test point in the domain, $b_0$ a length scale (0.88 km in the tests), $c$ the amplitude of the random noise (0.5 in the tests) and $a_t$ and $\varepsilon_{\boldsymbol{r},t}$ are random variables with normal distribution $N(0,1)$. It is assumed that idealised innovations are not correlated in time. For these data it is easy to prove that the covariance function at $\boldsymbol{r}_0$ is exactly the Gaussian curve $\Psi(\boldsymbol{r}) = e^{|r - r_0|^2 / 2b_0^2}$:

$$\overline{d_s(\boldsymbol{r}_0, t) d_s(\boldsymbol{r}, t)} = \overline{a_t^2 e^{|r - r_0|^2 / 2b_0^2} + a_t \varepsilon_{\boldsymbol{r}_0, t} e^{\|r - r_0\|^2 / 2b_0^2} + a_t \varepsilon_{\boldsymbol{r}, t} + c^2 \varepsilon_{\boldsymbol{r}, t} \varepsilon_{\boldsymbol{r}_0, t}}. \tag{21}$$

By using the facts that the average is linear, that the random variables are uncorrelated, and that $\overline{a_t^2} = 1$ we get:

$$\overline{d_s(\boldsymbol{r}_0, t) d_s(\boldsymbol{r}, t)} = e^{\|r - r_0\|^2 / 2b_0^2}, \quad \text{for } \boldsymbol{r} \in \Omega - \{\boldsymbol{r}_0\}. \tag{22}$$

For this exercise we consider only 2D "horizontal" isotropic covariance functions, which can be understood as a function along a constant $z$, $\sigma$, isopycnal or any other convenient computational surface. We tested these innovations at ten locations which cover areas with different characteristics (density of data and presence of land masses), as shown in Figure 2 on page 9. This type of innovation has only Gaussian covariance $\Psi(\boldsymbol{r})$ at $\boldsymbol{r}_0$, so the tests are repeated by making $\boldsymbol{r}_0$ equal to each location in the figure and constructing a different set of innovation data. The locations considered are a subset of nodes of a 2D grid of grid size $0.3° \times 0.275°$ (lon × lat) covering the area between longitudes $45°$–$74°$ and latitudes $8°$–$32°$. This grid was chosen for the simplicity of calculations and is different from the NEMO grid from which the innovations were computed. The benefit of using this synthetic innovation data is that the true covariance is perfectly known a priori, so it is possible to provide accurate measurements of the skills of the method. Since actual locations and times or real innovations were used, most of the problems associated to computing the covariance for real data were taken into account.

We used innovation data for three summer months (JJA), replicating an attempt of computing seasonal covariances. In total, we used nearly five million data points scattered over the whole domain, covering most of its surface with variable density of data. This amounts to an average of more than 50,000 data points per day in the domain.

For each location we also tested the sensitivity of having reduced amounts of data by randomly selecting different percentages of the innovation points. Tests were run for 100% (all the innovation points included), 75, 50, 20, 10, 5, 2 and 1%, where each test (or realisation) used different selection of innovation points. Most of the tests where

performed 30 times, with different random innovations, but the last three of 5, 2 and 1% were performed 60 times accounting for the larger statistical variability of the results.

Average number of innovations per grid cell per day

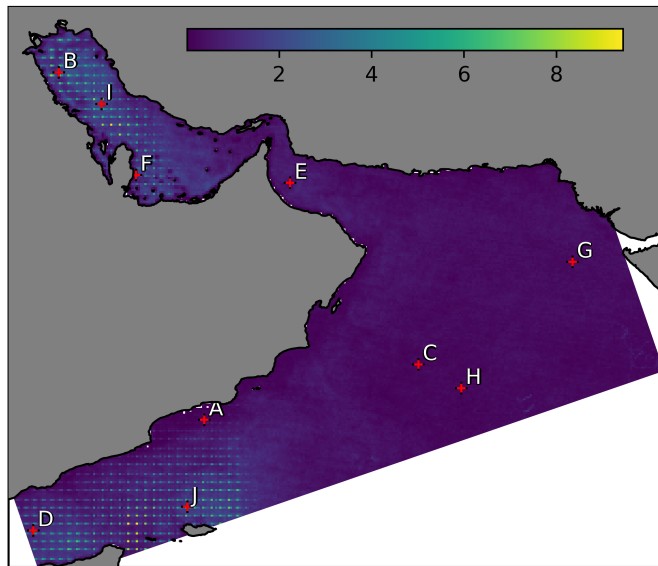

**Figure 2.** Domain of the operational system, showing the locations where the covariance functions were computed. The background colours show the density of innovations as the mean number of innovations per day on each grid cell.

In Figure 3 on page 10 we have plotted the results of the Gaussian curve $\Psi(r) = a\,e^{-(r-r_0)^2/2(88.8)^2}$ fitted to one realisation of innovation products using 100% of the data points. This is a 1D illustration of the actual 2D fitting process, but valid being a case of isotropic covariance. We have chosen locations B and H ($r_0 = r_B$ and $r_0 = r_H$) as being representative of how the fitting mechanism works. In the plots, the green dots are all pairs of products relevant to each location and the continuous black line is the Gaussian curve fitted to all these products using the PROM method. As it can be seen in Figure 2 on page 9, location B is in a region where there is high density of innovation data, while location H is in a region of relatively scarce amounts of data. This is mirrored in the density of innovation products in each plot of Figure 3 on page 10. It is worth noticing that the fitted line gives good results of the fitting parameter $a$, even if the green points seem to be scattered all over the place. For location B, where there is a large amount of data, the green dots cover densely a region that seems to be between two vaguely defined Gaussian curves, with the fitted function at an intermediate location. For H, located in an area with more scarce data, both in time and space, there are regions mostly empty of green points, but the fitting function still approximates quite well the covariance. The black dot at distance 0 is the variance of the innovations close to $r_0$ (i.e., in the central bin), which in this idealised case should tend to $1 + c^2 = 1.25$, where $c$ is the amplitude of the random noise as given by Equation (20). In most cases we tested, the black dots are above the value of the Gaussian curve at $r_0$. The difference between the computed variance and the value of the fitted curve at $r_0$ is the variance of the observations, which corresponds to the diagonal element of **R**.

Figure 4 on page 10 shows the errors of the fitting coefficient $a$ (the difference between it and the true value of 1), for all the realisations for the same points B, H and different amounts of data. The crosses (+) are the average values of the variance and the lines represent the standard deviation based on many realisations of randomly generated innovations. As one would expect, the mean errors are closer to zero when the number of data points is higher, as it is the variation of these errors in different realisations of the experiment. When the number of data points is very low, there are cases where the method is not able to produce a solution for the fitting coefficient and even when it is able, the error

can be quite high. However, there is a wide range of density of observations where the PROM method produces consistent results. This suggests that PROM is quite stable to changes in density of innovations.

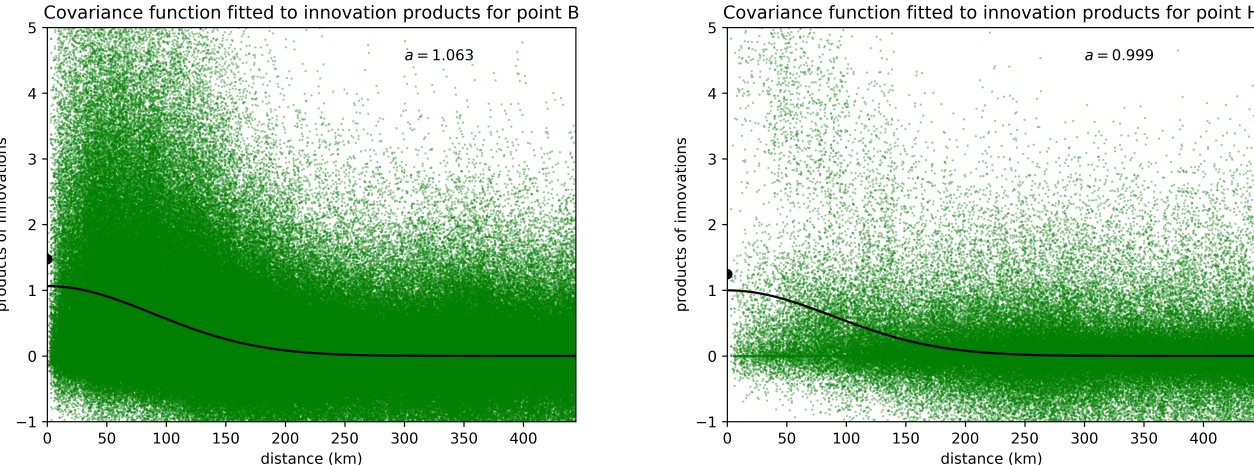

**Figure 3.** Gaussian function $\Psi(\boldsymbol{r}) = a\,e^{-(r-r_0)^2/2(88.8)^2}$ fitted to all pairs of innovation products around points B and H for one particular realisation and 100% of the data. The green dots are products of innovations plotted against the distance between them and the black line is the Gaussian fitted to all points using the PROM method. Each plot has a label with the fitting coefficient $a$, the theoretical value for this coefficient is 1. The black dots represent the variances at $\boldsymbol{r}_0$. The vertical axis limits have been chosen to make the figure more clear, about 3% of the green points lie outside of this range and they are not shown.

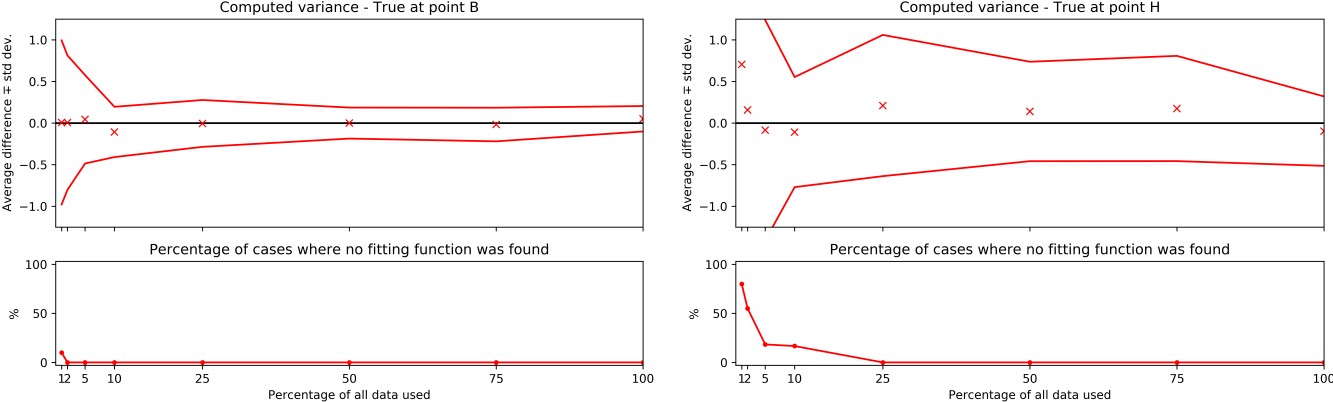

**Figure 4.** Sensitivity of PROM to the density of data. The top of each plot represents the error between the computed variance and the true value of 1 averaged for many realisations. The lines represent the standard deviation of these errors around the average. The bottom part of each plot shows how many of the realisations PROM method failed to obtain a result.

### 3.2. Real Case

The next set of experiments were designed to test the skill of the PROM method with real data. This is a more difficult task, since the covariance function is not known and hence, it is impossible to compute exactly the errors associated to each calculation of the covariance curve.

However, it is possible to test the validity of results in a different way that allow us to assess the skill of the method. For calculations we used the same summer season data from previous section, but in this case we used the actual innovations as computed by the NEMO operational model of the North Indian Sea. As pointed out before, we are assuming

ergodicity of time averages and a seasonal uniformity of the statistical properties of the temperature field. Following standard practices [12], the covariance function is assumed to be isotropic and that it can be modelled by the sum of two horizontal Gaussian functions accounting for a small scale (of the order of the Rossby Radius) and a large scale (a fixed, predetermined length scale) [18]. In other words, the covariance function can be written as:

$$\tilde{f}_{r_0}(\Delta r) = a_0 \, e^{\|\Delta r\|^2 / 2[L_r(r_0)]^2} + a_1 \, e^{\|\Delta r\|^2 / 2L^2}, \tag{23}$$

where $L_r(r_0)$ is the Rossby Radius at $r_0$ (capped from below to 25 km), $L$ is the large length scale fixed at 444 km and $r_0$ is the location of each node in the grid. The free parameters $a_0$ and $a_1$ were obtained by the PROM method. This choice of two Gaussian functions of different length scales is adopted from the literature [18] as it has shown good correspondence with real background error covariance functions, including the local effects due to ocean circulation and the regional effects of larger scale.

Figure 5 on page 11 shows the fitting of $\tilde{f}_{r_0}(\Delta r)$ to real innovation products at points B and H. As with the idealised case, there are products of innovation pairs scattered all over the place (green dots), but the PROM method was able to fit the covariance model to these data.

To study the sensitivity of PROM to data density, we also run many tests for different percentages of the total data. For each test, only a percentage of all innovations were randomly selected (100% meant that all available innovations were used). As for the ideal case, we repeated the test many times for each percentage, where each test (or realisation) differs from another only on the percentage of innovations and which innovations were selected. This contrasts with the idealised case, where the innovations also changed their values randomly.

Figure 6 on page 12 shows the coefficients $a_0$ and $a_1$ computed using PROM for different percentages of innovation data for the same two locations B and H. PROM produces results of the average variance that are stable in a wide range of data densities. For example, for location B, the estimated variance using all the innovation data was $0.058\,^{\circ}\mathrm{C}^2$ and for 25% of the data, $0.063\,^{\circ}\mathrm{C}^2$. For location H, the estimated variance for 100% was $0.028\,^{\circ}\mathrm{C}^2$ and for 25% of the data, $0.027\,^{\circ}\mathrm{C}^2$.

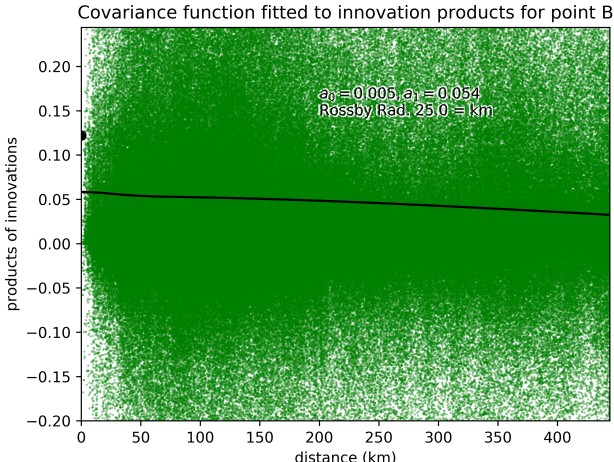
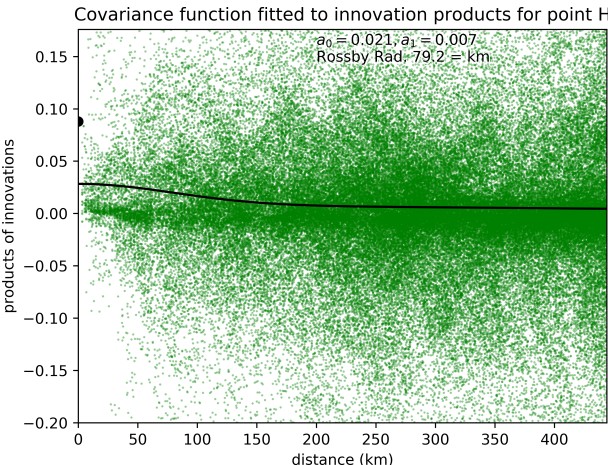

**Figure 5.** Linear combination of Gaussian functions, $\Psi_2(r) = a_0 \, e^{\|r - r_0\|^2 / 2[L_r(r_0)]^2} + a_1 \, e^{\|r - r_0\|^2 / 2L^2}$ fitted to all pairs of innovation products around points B and H for 100% of the data. The green dots are products of innovations located at the distance between them, and the black line is the function fitted to all points using the PROM method. The black dot represents the variance at $r_0$. The values of $a_0$ and $a_1$ and the Rossby Radius are shown in labels on each plot. The vertical axis limits were chosen to make the figure more clear, about 3% of the green points lie outside of this range and they are not shown.

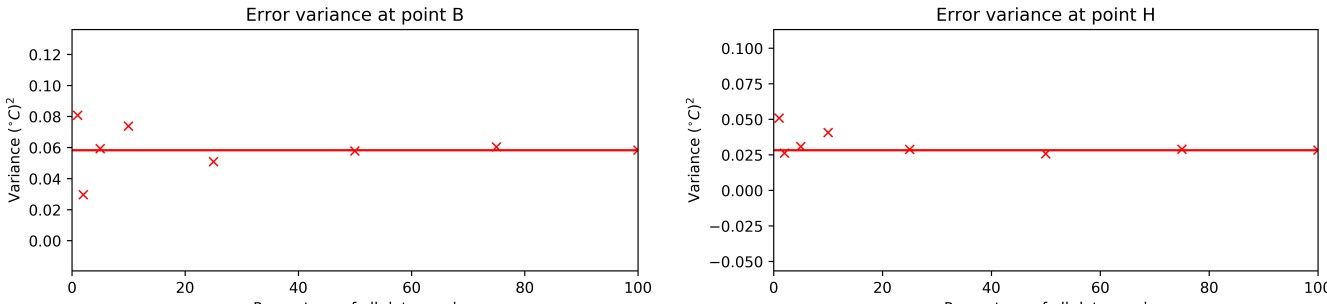

**Figure 6.** Sensitivity of PROM to the density of data. Each plot represents the computed error variance averaged for many realisations of randomly selected data points. The number of realisations for which PROM method failed to obtain a result are the same as in the idealised case (see Figure 4 on page 10).

Figure 9 on page 17 and Figure 10 show the variances and length scale ratios computed using PROM for 100% of the data. The method was able to produce results everywhere in the domain, even in the regions where the amount of data is scarce. Further analysis is given in the Discussion section.

As the true solution for the covariance function is not known, objective estimations on the quality of the results are obtained by indirect methods. The number of points where the method is not able to compute the covariance function is one of these objective measurements. Another test that can be applied to the grid points where is based on the Cauchy–Schwartz inequality applied to covariances. Let us consider two random variables $\alpha$, $\beta$, the Cauchy–Schwartz inequality states:

$$\mathrm{Covar}(\alpha, \beta)^2 \leq \mathrm{Var}(\alpha)\mathrm{Var}(\beta). \tag{24}$$

This expression is always true, in particular, it can be applied to the innovations at any pair of locations in the domain $r_1$, $r_2$ as $\mathrm{Covar}(d(r_1), d(r_2))^2 \leq \mathrm{Var}(d(r_1))\mathrm{Var}(d(r_2))$. In reality, we do not know the covariances and variances exactly, only the approximate covariance functions $\tilde{f}_{r_1}$, $\tilde{f}_{r_2}$. For these functions, the inequality does not hold in general, but we can use it as a test. If $\tilde{f}_{r_1}$ and $\tilde{f}_{r_2}$ do not satisfy the inequality, they cannot be real covariance functions. One must be careful because many methods of calculating error covariances (PROM and H-L included) produce inconsistent covariances in the sense that, in general, for two different points $r_1, r_2, \tilde{f}_{r_1}(r_2 - r_1) \neq \tilde{f}_{r_2}(r_1 - r_2)$ (this problem is usually fixed at a later stage, see, for example, in Weaver and Courtier [23] how covariance matrices are made symmetric). To avoid this problem we compute the approximate covariance as the minimum value between the two values provided by $f_{r_1}$ and $f_{r_2}$:

$$\min\left(\tilde{f}_{r_1}(r_2 - r_1), \tilde{f}_{r_2}(r_1 - r_2)\right)^2 \leq \tilde{f}_{r_1}(\mathbf{0})\tilde{f}_{r_2}(\mathbf{0}) \tag{25}$$

For this test, we compared all nodes with the node to its immediate one the east using Equation (25) and plotted a yellow cross in the location of the nodes that do not satisfy it. These results can be seen in Figure 11 on page 18, where it can be seen that the concentration of "uncertain" grid points (i.e., those which do not satisfy the condition) is higher in regions of lower density of data or near the coast.

## 4. Discussion

The PROM method shares points in common with the H-L method and for this reason, most of the discussion consists of comparisons between both methods. For H-L we used a standard implementation of the method based on the initial description by Hollingsworth and Lönnberg [13]. Different groups may use different implementations which can differ in a few technical aspects. We have checked that the implementation of the H-L method used for comparison in this section reflects standard practices used by the MetOffice (private

communication). Briefly, our implementation of the H-L method consists of the following steps. First, all possible products of pairs of innovations relevant to each model grid node, are calculated and binned by their distances. Second, the products related to a certain bin are averaged (usually using time averaging) to give an approximate value of covariance. Third, a curve is fitted through the covariance points (not including the first bin) to produce a covariance function which then is used to calculate the diagonal element of **R** related to that grid node, and a row of elements for the **B** matrix, both diagonal and non-diagonal. The central bin around the grid point in question in the H-L has the same size as in PROM method.

A bin is considered valid if it contains products of innovations coming from, at least, $K$ different time intervals of length $\Delta t$ (one day in our case). Otherwise, it is assumed that there are not enough data to compute the covariance in an statistically meaningful way and the bin is disregarded. This is done, instead of just putting a limit on the total amount of products of innovations per bin, to avoid cases where there are relative large amounts of data in a bin, but coming from just a few (or even just one) time intervals. In cases like this, all the innovations in the bin come from few different ocean states and the statistics cannot be trusted. In this section we present results for $K = 5, 10$ and $20$. Additionally, as we did in PROM, if the central bin is empty for one time interval, the data for the whole interval are also disregarded. The results obtained by applying the H-L method are then compared with our PROM results.

Usually, the covariance matrices are not used directly as computed by methods like H-L. In practice, a post-processing stage deals with missing values and noise. We will not discuss these methods as they would be similar for any method, and will focus only on the raw results.

### 4.1. Idealised Case

For the idealised case we only considered the results at locations A–J (see Figure 2 on page 9). For the locations with a small number of available data, H-L often failed to produce results. Sometimes because there were not enough data for the fitting procedure, other times because the results obtained by H-L for the variance exceeded a prescribed threshold $|\text{variance} - 1| > 10$. In contrast, PROM produced results much more often, even in those cases where H-L fails. This is represented in the lower part of each plot in Figure 7 on page 15, where the blue lines correspond to the percentage of cases when the H-L method was not able to find a valid value for the variance and the dashed and the red line does the same for PROM. As it can be seen, the H-L method completely failed to produce results for locations A, C and H where data density is lower, and only produced results for high percentage of the data in the rest of the points. On the other hand, PROM produced results in all points when 100% of the data was included. In most cases, this happened even for percentages below 10%.

Finally, PROM usually produces better results to H-L. Not only in terms of the value of the variance averaged among all realisations, but also showing less variation of those values. Moreover, the method is more stable as the number of available data decreases, showing good skill in the whole range shown.

The results obtained are summarised in Figure 7 on page 15, where for all the locations and selected percentages we show the average error (difference between the computed variance and the true value of 1) and the range of the standard deviation of errors computed from all realisations.

### 4.2. Real Case

For the real case we produced results in the whole domain, however, firstly, we will consider only the results at points A–J to see how they compare. Additionally, we can use the knowledge obtained from the idealised case to estimate the quality of the results. In this case, the term "realisation" means the subset of innovations which are selected randomly from the full data set. For a particular percentage (for example 50%), different

"realisations" contain 50% data of the full data set, but the set of innovations chosen is different in each case. The process of selection of innovations used the same random seed as in the idealised case and, for this reason, the number of cases where both methods failed is identical to the idealised case (see Figure 7 on page 15) being much more likely that H-L fails compared to PROM. Figure 8 on page 17 shows the variances obtained for all the percentages and locations in Figure 2 on page 9 averaged for all realisations. There are many more cases where PROM is able to produce a result compared to H-L. The variance results produced for 100% of the data are "similar" or, at least, not entirely dissimilar, for both methods, being them usually more similar for locations where the density of data is higher. In general, PROM stays close to its horizontal line (red) for a wider range of data amount percentages than H-L does to its line (dash blue). In practice this means that PROM method will produce valid covariance functions much more often than H-L.

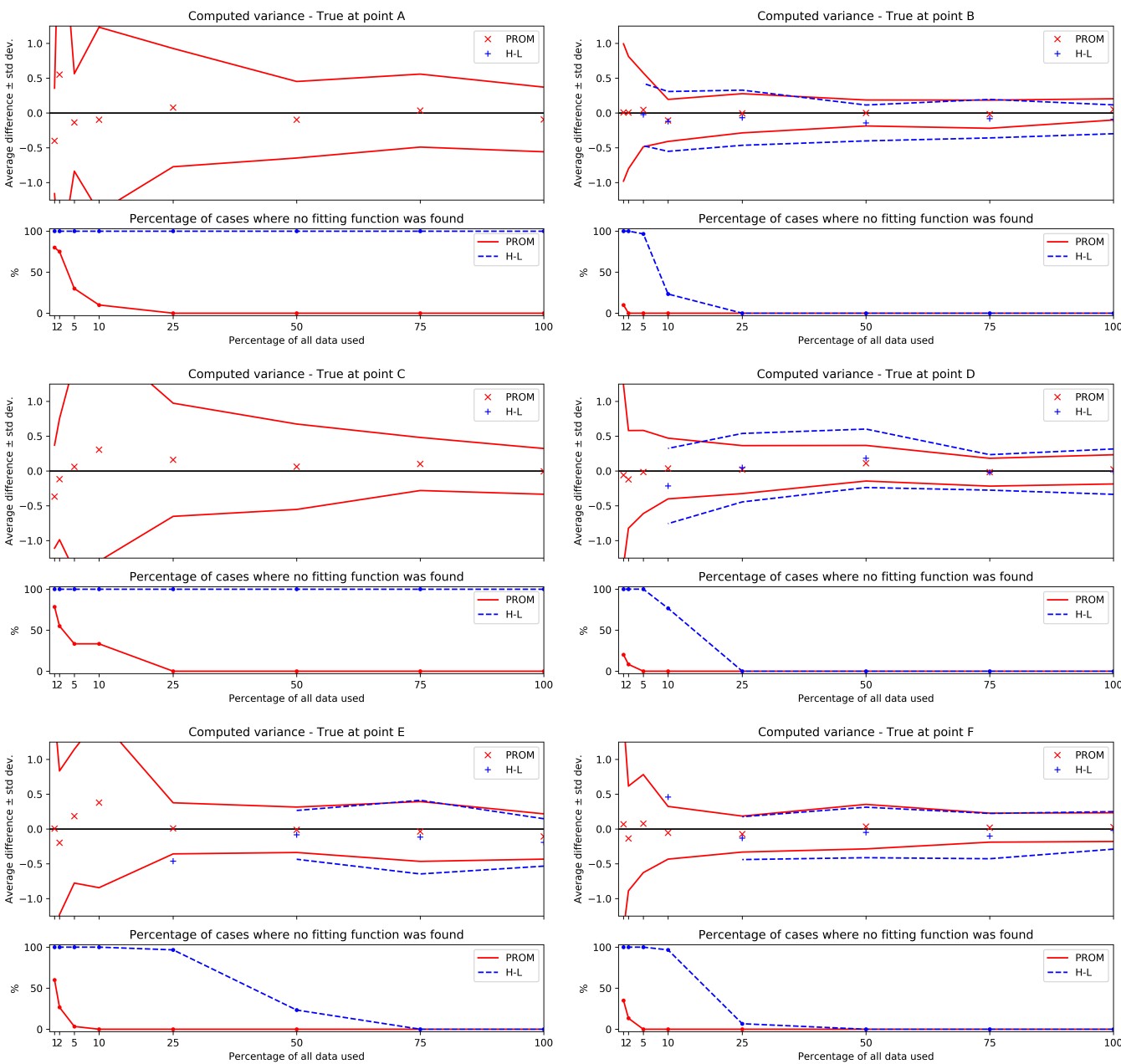

**Figure 7.** *Cont.*

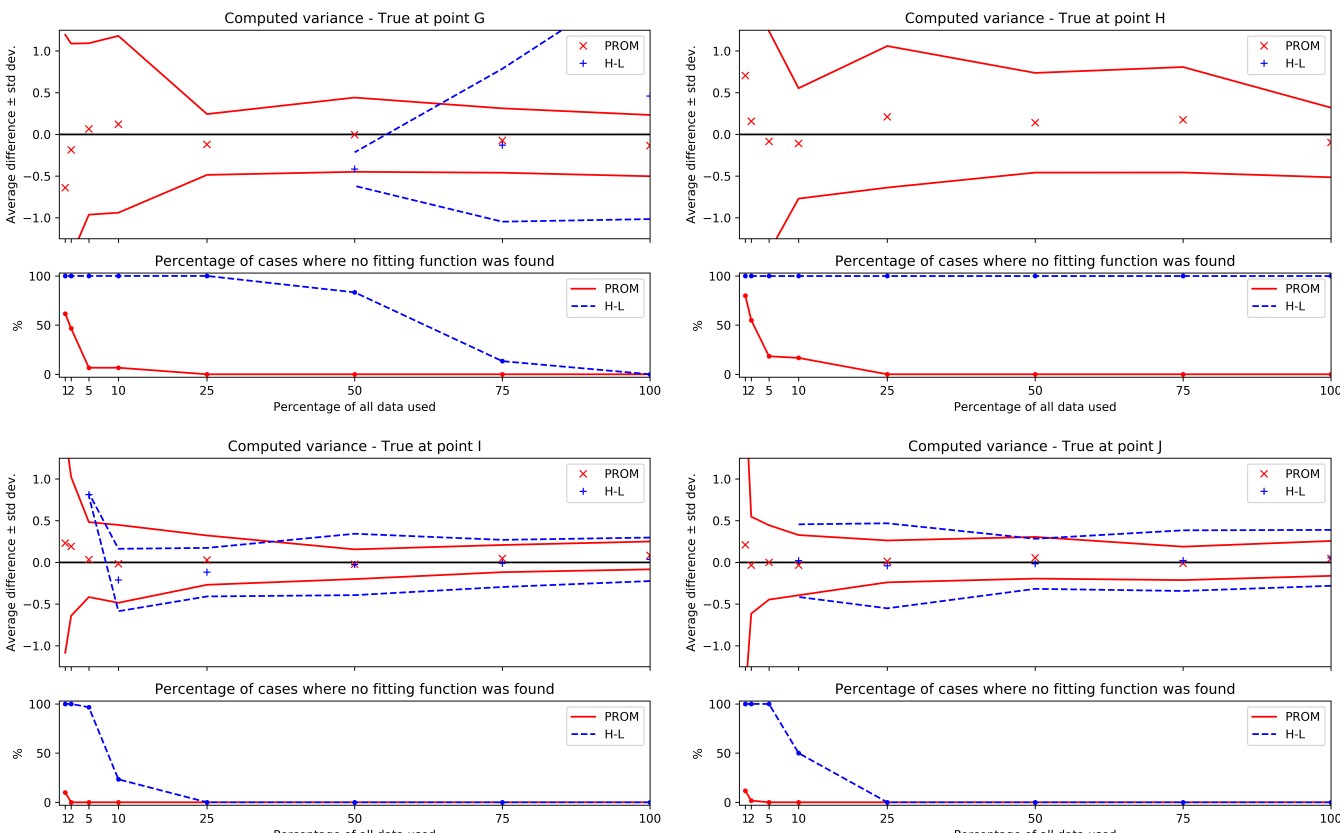

**Figure 7.** Comparison of variance values obtained by H-L and PROM methods. Each plot represents results obtained from at least 30 numerical experiments realised at each point (see Figure 2 on page 9). The top part of each plot shows comparisons of differences of the computed variance with respect to the true value of 1 (red × for PROM method, blue + for H-L). The lines represent the standard deviation of computed values for all realisations of the same experiment with different random innovations. The bottom part of each plot shows the percentage of cases where each method failed to produce a value of the variance.

The results for the variance for the entire domain computed using PROM and H-L are shown in Figure 9 on page 17, where we have included results of H-L for different minimum number of days required per bin ($K = 5, 10, 20$).

Even for the rather small value of $K = 5$, H-L was unable to produce results in the some parts of the domain, while PROM was able to find a result of the covariance function for all grid nodes. This limitation of H-L can be mitigated somewhat by increasing the size of the central bin so there are more innovations and hence more product pairs, however, there are two things to consider. First, increasing the size of the central bin reduces the spatial resolution of the method and requires to stretch the condition of local homogeneity to larger scales; second, the results obtained this way should be compared with PROM results obtained with the same enlarged central points. In this case, PROM will also have more data to work with and the covariance results will improve as well.

In this real case, we do not know the true error variances, but by inspecting Figure 9 on page 17 and Figure 10 one can extract some conclusions about the quality of the results obtained. The modelled error variances and length scales obtained from H-L are slightly noisier than those obtained using PROM, even in the regions where data are less scarce. In these areas, the results of H-L and PROM are very similar, which means that when there are plenty of data, both methods converge to the same results. In the regions with smaller data density, H-L often fails to produce a result for the covariance.

The plots in Figure 11 on page 18 show that the number of "uncertain" points (those which do not satisfy Equation (25)) is much smaller for PROM (779) than for H-L (1622). Not surprisingly, the uncertain points are more common near the coast lines and in the

regions where the density of innovations is low (see Figure 2 on page 9). Once again, PROM seems to be able to extract more information from the available data.

The PROM method can also be much faster than H-L. This is possible because PROM can be written in terms of convolutions of the basis functions $\phi_i$ with the covariance function. In the case when $\phi_i$ are separable (they can be written as products of functions depending only on each spatial coordinate) as Gaussian functions are, the convolution can be computed with reduced computational complexity. This is described in detail in Appendix A where computational complexity estimates are also given.

Our Python implementation of both methods was optimised using the JIT compilation library NUMBA [24] to increase the performance of two inner loop functions in the case of H-L and using the method described in Appendix A for PROM when computing the maps. The computing time for a typical case of 100% data was 27 s for PROM and 300 s for H-L. A very sizeable computational time reduction consistent with the time computational complexities is shown in Appendix A.

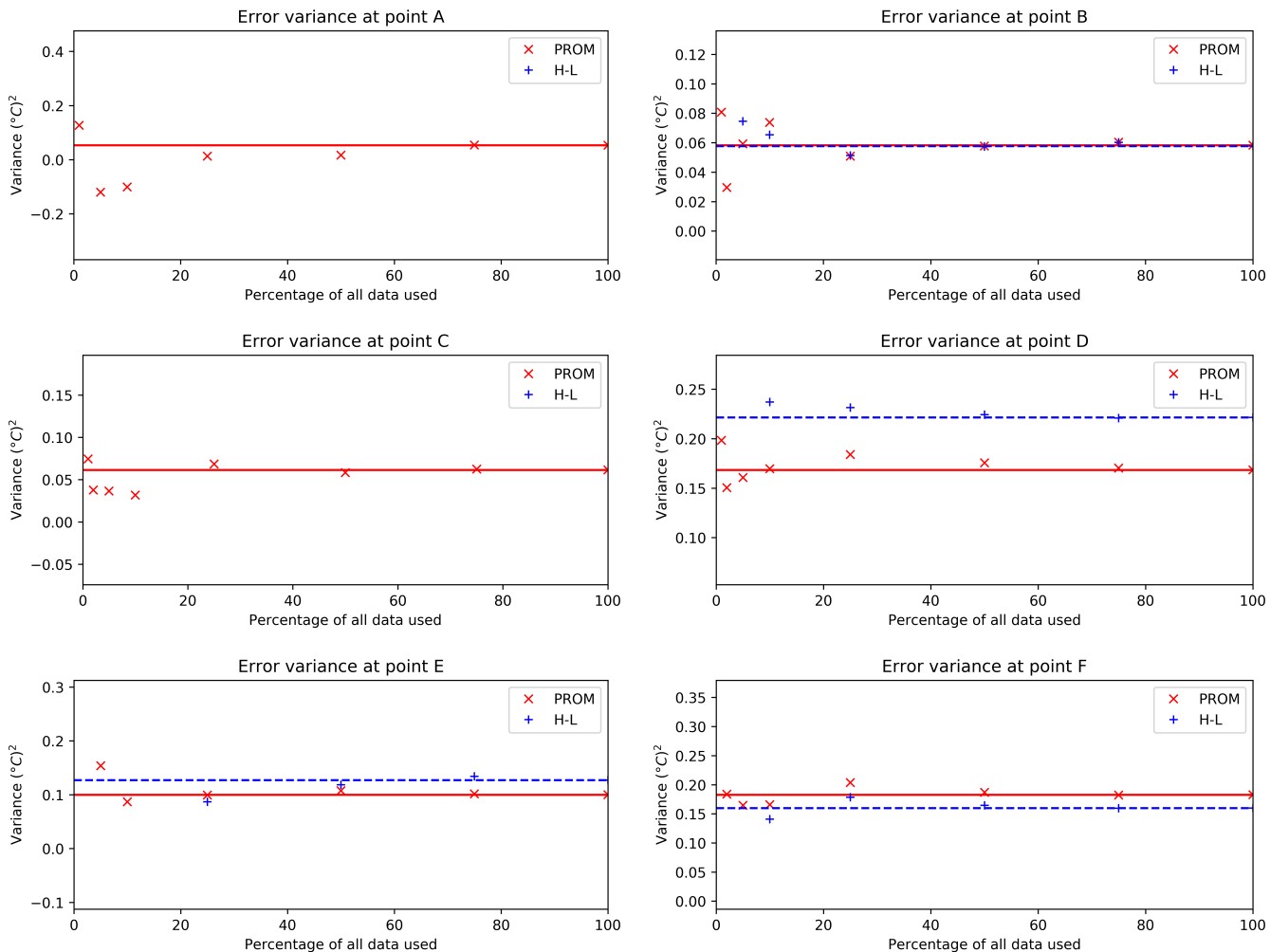

**Figure 8.** *Cont.*

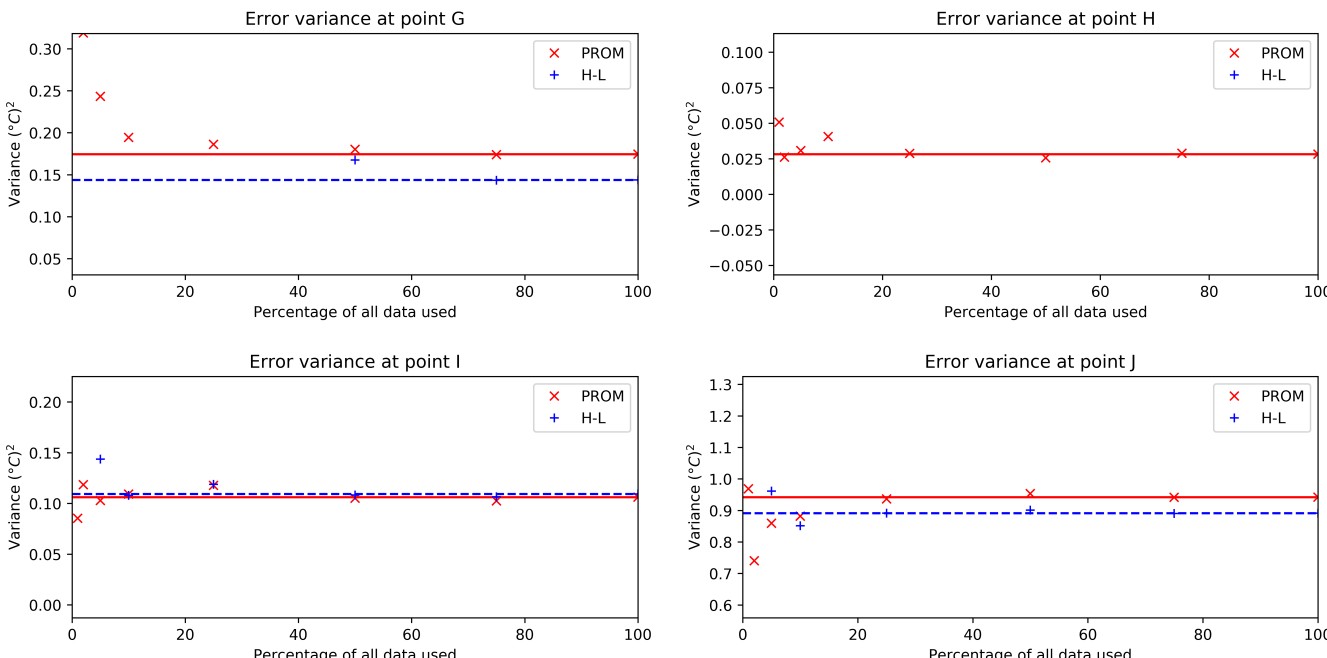

**Figure 8.** Comparison of H-L and PROM methods. Each plot represents results obtained from many numerical experiments realised at each location (see Figure 2 on page 9) showing comparisons of the computed variance (red × for PROM method, blue + for H-L). The respective horizontal lines are located at the value of the variance obtained for 100% in each method.

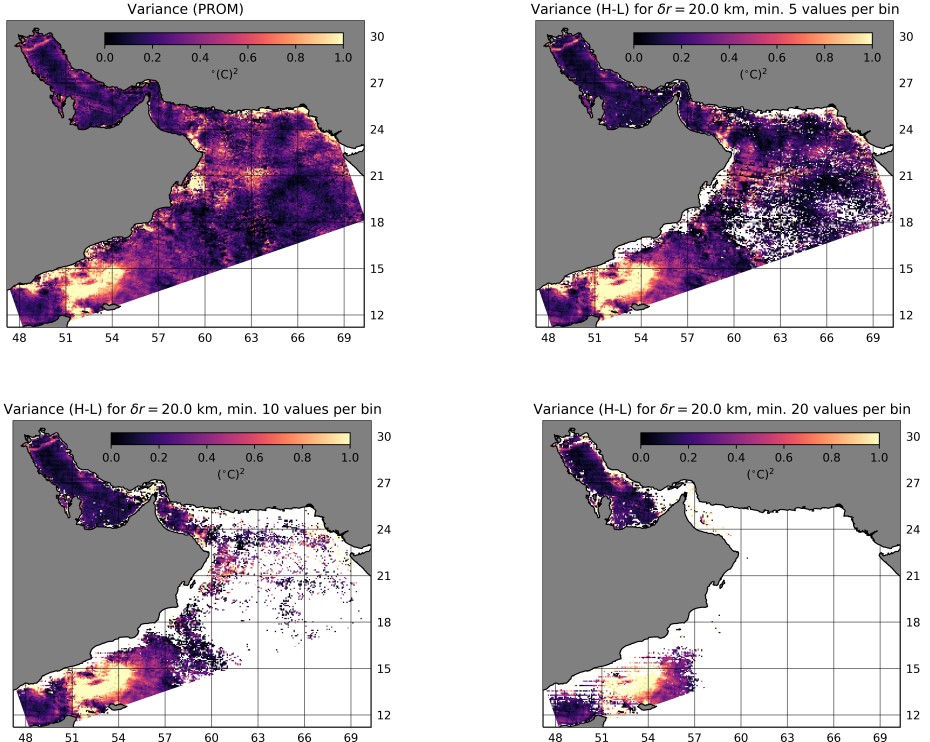

**Figure 9.** Background model error variances ($a_0 + a_1$) computed using both methods. For H-L we have included results for bin sizes $\delta r = 20$ m and different minimum number of days to consider a bin valid for computing covariances ($K = 5, 10, 20$). H-L method is very sensitive to K, and only produces results in most of the domain nodes for $K = 5$.

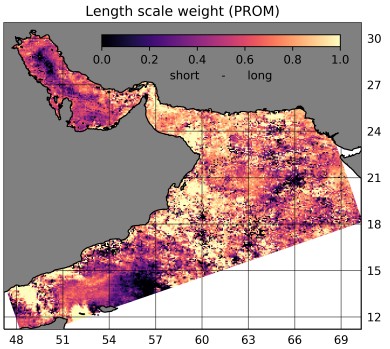 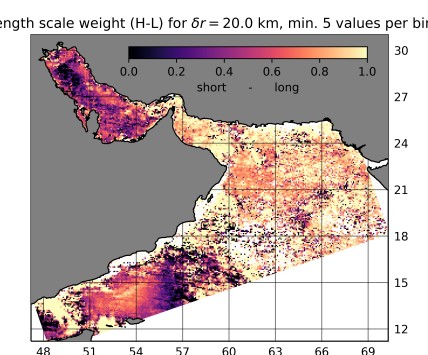

**Figure 10.** Weights of the length scales of the background model error ($a_1/(a_0+a_1)$) computed using both methods. For H-L we used $\delta r = 20$ m and a minimum of days per bin, $K = 5$.

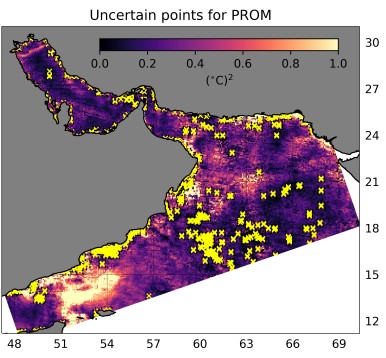 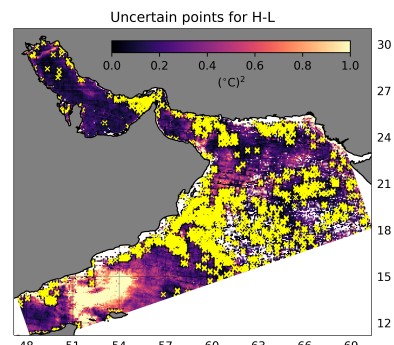

**Figure 11.** Points that do not satisfy the inequality in Equation (25) for horizontally adjacent valid grid nodes. In the left plot are the 779 uncertain points found for the PROM method, in the right, the 1622 uncertain points found for H-L with a bin size of 20 km and a minimum number of days per bin $K = 5$.

## 5. Conclusions

We have presented a novel method to compute the error covariance matrices based on analysis of innovations named Projection Method (PROM). Similarly to the widely used H-L method [13], PROM uses statistics of innovations, i.e., differences between model and observations. The main novelty is that PROM does not require spatial binning of the innovations and all the statistics are performed over the whole available data set. This means that no innovation value is discarded which allows to obtain better results by making better use of all the information. It also means that the method is simpler to implement, especially for anisotropic cases.

To test the new method we first applied it to an idealised case, where the location and timings of the innovations are real (coming from an operational model of the North Indian Sea), but the innovation values are substituted by a synthetic field based on random fluctuations following a known covariance function. Then, PROM was applied to real data from the same modelling system.

The idealised case demonstrates that the PROM method performs better than H-L when compared to a known, true covariance function. The results obtained in ten selected geographical locations and for different amounts of input data are used to test the sensitivity of each method to the data available. The PROM method is shown to produce reasonable results even when the H-L method fails due to insufficient amount of data. In areas with large number of observations both methods produce similar results.

In the case of real data for innovations, the true solution for the covariance function is not known, so the capabilities of both methods are assessed indirectly. In regions with low density of observations, the H-L method often fails to produce a valid result, especially

for some choices of the bin sizes ($\delta r$) and minimum number time intervals per bin ($K$). On the other hand, PROM always produced a result. The test based on initial description inequality for covariances shows that the H-L method results fails the test in more than twice the number of points of grid nodes as compared with PROM.

Finally, we developed a computational optimisation scheme, which is based on the concept of "separable convolution", that can be applied to our method but not to H-L. This optimisation reduces the algorithmic complexity of the method and is able to reduce computation times greatly. Both methods were implemented in Python, and PROM computational time is seven or more times shorter than H-L using similar levels of code optimisation.

**Author Contributions:** Conceptualisation, J.M.G.-O., L.S and G.I.S.; methodology, J.M.G.-O. and L.S.; software, J.M.G.-O. and L.S.; writing, J.M.G.-O., G.I.S. and L.S.; supervision, G.I.S. All authors have read and agreed to the published version of the manuscript.

**Funding:** This research received no external funding.

**Institutional Review Board Statement:** Not applicable.

**Informed Consent Statement:** Not applicable.

**Conflicts of Interest:** The authors declare no conflict of interest.

## Appendix A. Optimisation of the Algorithm

PROM method is conceptually simpler than the H-L method because binning is not necessary nor is there the need to compute each bin's covariances. However, in terms of computational complexity both methods are equivalent when the trivial implementations are used. Let us assume for simplicity that we want to compute the covariance functions for a square mesh of $M \times M$ nodes and that, for each node, there are an average of $n$ innovations inside its central bin, and $N$ other innovations considered to form innovation pairs. With this assumptions the computational complexity will be $O(nNM^2)$ (number of nodes times the number of pairs) for both methods when using the trivial algorithm. Usually, $N$ is a big number (typically, hundreds of thousands) so the time spent in the calculations can grow very large as $M$ increases. Not requiring as many steps as H-L, PROM is simpler and the implementation can potentially be faster than H-L, but not by much.

However, PROM has one advantage that allows to reduce the algorithmic complexity to just $O(MT)$, which can result in huge time savings when computing large covariance matrices. Let us call $\boldsymbol{r}_{pq}$ the location of a grid node of the output grid with indices $(p, q)$. The inner products that need to be computed for each node are (see Equation (19)):

$$\left\langle f_{\boldsymbol{r}_{pq}}, \phi \right\rangle \approx \sum_{i=1}^{N} d_0(t_i) d(\boldsymbol{r}_i, t_i) \phi(\Delta \boldsymbol{r}_i), \tag{A1}$$

where we have dropped the error term $O\left(N^{-1/2}\right)$ and opted for a more concise notation. Let us now suppose that instead of basis functions $\phi$ we use discretised versions of them $\phi'$, such that $\phi\left(\boldsymbol{r}_{pq}\right) = \phi'\left(\boldsymbol{r}_{pq}\right)$, and that the values of $\phi'$ are constant inside each cell of the grid with centres at $\boldsymbol{r}_{pq}$. In other words, $\phi'$ are discretised, non-continuous, stair-like versions of $\phi$ functions. In this case, Equation (A1) can be rewritten as:

$$\left\langle f_{\boldsymbol{r}_{pq}}, \phi' \right\rangle \approx \sum_{l=1}^{M} \sum_{m=1}^{M} \sum_{t=1}^{T} d_0(t) D_{lm}(t) \phi(\Delta \boldsymbol{r}_{lm}), \tag{A2}$$

where $D_{lm}(t)$ are the sums of all the innovations inside each grid cell $(l, m)$ at time index $t$. Equation (A2) is now formally a 2D spatial convolution that can be computed in $\mathcal{O}(MT)$ if $\phi'$ is separable, which is the case for 2D Gaussian functions and their discretised versions. Pre-computing the required $d_0(t)$ and $D_{lm}(t)$ takes a comparatively small amount of time

and we will not take it into account here. In most cases, $MT$ is much smaller than $nNM^2$ and hence PROM can be much faster than H-L. This was confirmed by the timings of our implementations for PROM and H-L methods.

One could argue that the discretisation of $\phi$ as $\phi'$ is another way of performing a spatial binning, and in some respects this is true. There are two main differences, however. First and more importantly, there is no statistical requirement on the minimum number of innovations inside each bin; second, the output grid can be the same mesh we are using in the ocean model, so function $\phi'$ is consistent with the rest of the variables (e.g., the SST) that are also discretised in a similar way.

At a first glance, the optimised code for PROM seems not well suited for non-homogeneous covariance functions like Equation (23). The reason is that the separable convolution method only works for homogeneous, separable functions. However, this limitation can be easily overcome by performing the inner product for Gaussian functions of several length scales in the range of values that take the Rossby radius in the area, and use these values to interpolate to the desired length scale.

In this case, we have performed convolutions with Gaussian curves of length scale equal to the large length scale ($L$) and for $L_r$ taking 20 values between 25 and 133 km (the minimum and maximum Rossby radius values considered). Even with the added cost of computing the extra convolutions, PROM is almost an order of magnitude faster than H-L. In this paper, all the results computed for the real data have been computed using the method presented in this appendix after testing that they were almost identical to those obtained using the trivial algorithm.

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
