# Peer review of "A Projection Method for the Estimation of Error Covariance Matrices for Variational Data Assimilation in Ocean Modelling"

_jmse, doi:10.3390/jmse9121461_

Round 1

Reviewer 1 Report

This paper on data assimilation proposed a new method to treat the covariance matrix. This Projection Method (PROM) has the potential to be a useful tool for data assimilation of ocean modelers. The authors have also tested the PROM using multiple application and the results look very good. The presented work is well-suited for JMSE journal and should be published.

I only have several minor concerns:

  1. The authors should give the full name of PROM in line 21.
  2. The authors should explain the selection of Arabian Gulf in this paper. Why is Arabian Gulf so important?
  3. I would recommend the authors show all the green points in Figs. 3 and 5.
  4. I think the authors should explain why PROM is running much faster than H-L. It is a major innovation of PROM, but the authors did not explain it.

Author Response

Comment: This paper on data assimilation proposed a new method to treat the covariance matrix. This Projection Method (PROM) has the potential to be a useful tool for data assimilation of ocean modelers. The authors have also tested the PROM using multiple application and the results look very good. The presented work is well-suited for JMSE journal and should be published.

Response: Thanks

Comment: The authors should give the full name of PROM in line 21.

Response: The proposed change has been added to the revised manuscript.

Comment: The authors should explain the selection of Arabian Gulf in this paper. Why is Arabian Gulf so important?

Response: The Arabian/Persian Gulf has a complex bathymetry and circulation pattern, and therefore, it is a good place to test the algorithm. The rationale for this choice has been made more clear in the revised manuscript.

Comment: I would recommend the authors show all the green points in Figs. 3 and 5.

Response: The green points come from products of innovations that can have extreme values and showing all of them will reduce the scale of the fitted Gaussian, so we decided to zoom in for clarity. The number of points not plotted is typically less than 3%.

In the revised manuscript we have included a note in the figure captions explaining that some green points have been clipped out to show the fitted function more clearly.

Comment: I think the authors should explain why PROM is running much faster than H-L. It is a major innovation of PROM, but the authors did not explain it.

Response: It is explained in the appendix. We have amended the main text to make it more clear.

Reviewer 2 Report

Review.

The article describes a practical method for estimating the covariance function for forecast/observational errors. Evaluation of this function is of great importance because it is used in numerical  hydrometeorological forecasts. This method can be considered as a simplification of the known Hollingsworth – Loennberg method. The article is written quite clearly, the results and conclusions are well substantiated and illustrated by the data of numerical experiments.

I recommend to accept it after minor revision (including to the following remarks).

  1. P. 8. “Hollingsworth and Lnnberg” – Loennberg.
  2. Check Formula (3) - different definitions of x_t?
  3. Formula (5) - define the line above. Is this time averaging?
  4. P. 168. "Once this is done, the fitting parameters ai will be computed using a norm minimization approach akin ? to the least ..." - correct.
  5. I advise the authors to carefully read and correct the text in paragraph 2.3, p. 171-191.

Here f is the Lebesgue function, which should be redefined on a subset of measure 0? Maybe we can do with the usual Hilbert space with the L_2 norm? For example, the space of piecewise continuous functions, functions with finite support, etc.?

  1. Refine in (9): f is a piecewise constant function, function with compact support or?
  2. F. (10). If f is a Lebesgue function, then where are the integrals of it? Or is f a nonzero function only in the vicinity of observation points?
  3. P. 208-210. "These types of equations are a very common way of finding the projection of an element of space into a subspace and very similar ideas can be found, for example, in the Petrov-Galerkin method of solving partial differential equations."

It is not clear what the PDE solution and the Petrov-Galerkin method have to do with it? Also, there is no link. If this text is retained, then it should be clarified.

7 (1). Since the RLH sides of (12) have the same functions, this is the Bubnov-Galerkin method (the basic and projection systems are the same). In the Petrov-Galerkin method the basic and projection systems are different. Add, please, the corresponding reference.  

7 (2). Take into account and write that the B-G method is a method for solving an operator equation in the form of a linear combination of basic elements, which is a linearly independent system!

7 (3). The problem of determining the coefficients from system (12) is not always equivalent to finding the minimum (11). In general, you need to prove it or at least assume.

7 (4). A system like (12) can be ill-conditioned. In this case, small errors on the right hand side f_r will lead to large changes in the solution a_1, a_2. Maybe this should be noted and discussed briefly. This is especially important in the case of high dimensionality!

Author Response

Comment: The article describes a practical method for estimating the covariance function for forecast/observational errors. Evaluation of this function is of great importance because it is used in numerical  hydrometeorological forecasts. This method can be considered as a simplification of the known Hollingsworth – Loennberg method. The article is written quite clearly, the results and conclusions are well substantiated and illustrated by the data of numerical experiments.

I recommend to accept it after minor revision (including to the following remarks).

Response: Thanks.

Comment: P. 8. “Hollingsworth and Lnnberg” – Loennberg.

Response: The typo has been corrected as advised.

Comment: Check Formula (3) - different definitions of x_t?

Response: The expressions in eq (3) are not a definition of x^t, but rather of the errors \eta and \epsilon. We have reordered the equations to avoid confusion.

Comment: Formula (5) - define the line above. Is this time averaging?

Response: It is ensemble average, which in practice is estimated using time average. In the revised manuscript we have defined the overline and made more clear its relation with the time average.

Comment: P. 168. "Once this is done, the fitting parameters ai will be computed using a norm minimization approach akin ? to the least ..." - correct.

Response: It is not clear to us what this comment means but we have substituted “akin” by “similar” as it is a more common word.

Comment: I advise the authors to carefully read and correct the text in paragraph 2.3, p. 171-191.
Here f is the Lebesgue function, which should be redefined on a subset of measure 0? Maybe we can do with the usual Hilbert space with the L_2 norm? For example, the space of piecewise continuous functions, functions with finite support, etc.?

Response: We are not claiming that f must me redefined, the norm proposed in equation 12 just takes the values at some discrete locations, but the function is still the same.

The reviewer’s suggestion that may be we could do with the usual Hilbert space is intriguing. But this will be a substantially different approach because it will require to modify function f which will require to modify several other technical details.

We believe that our approach of defining the norm is simpler, however, we have rewritten parts of the section to remove the reference to the usual L2 norm and Lebesgue. It was given just as an illustration, but the reviewer’s comment suggest that it might be, in fact, confusing.

Comment: Refine in (9): f is a piecewise constant function, function with compact support or?

Response: No, as we have mentioned before fr is unchanged. We think we have made the derivation confusing by introducing the Lebesgue integral, which may suggest that eq. (9) is derived from the previous one when, in fact, it is a new definition.

fcan be continuous and does not require to have compact support. Using the proposed discrete norm, these considerations are not required.

Comment: F. (10). If f is a Lebesgue function, then where are the integrals of it? Or is f a nonzero function only in the vicinity of observation points?

Response: Function f is not necessarily Lebesgue integrable since the norm defined in (9) does not require it to be integrable at all. The integrals were only included to show that our norm definition is not dissimilar from the Lebesgue one. We have reformulated this section to make it more clear avoiding any reference to Lebesgue.

Comment: P. 208-210. "These types of equations are a very common way of finding the projection of an element of space into a subspace and very similar ideas can be found, for example, in the Petrov-Galerkin method of solving partial differential equations."
It is not clear what the PDE solution and the Petrov-Galerkin method have to do with it? Also, there is no link. If this text is retained, then it should be clarified.

Response: Thanks. Taking into account the reviewer’s comments, we have decided to remove references to the Galerkin methods and opted for a more direct derivation.